# Response of Pasture Nitrogen Fertilization on Greenhouse Gas Emission and Net Protein Contribution of Nellore Young Bulls

**DOI:** 10.3390/ani12223173

**Published:** 2022-11-16

**Authors:** Lais Lima, Fernando Ongaratto, Marcia Fernandes, Abmael Cardoso, Josiane Lage, Luis Silva, Ricardo Reis, Euclides Malheiros

**Affiliations:** 1Department of Animal Sciences, University of Florida, Gainesville, FL 32611, USA; 2Department of Animal Sciences, Sao Paulo State University, Jaboticabal 14884-900, SP, Brazil; 3Range Cattle Research and Education Center, University of Florida, Ona, FL 33865, USA; 4Cargill Animal Nutrition, Campinas 13091-611, SP, Brazil; 5Queensland Alliance for Agriculture and Food Innovation, The University of Queensland, Brisbane, QLD 4072, Australia; 6Department of Mathematics and Statistics, Sao Paulo State University, Jaboticabal 14884-900, SP, Brazil

**Keywords:** feed conversion, greenhouse gases, human-edible protein, sustainability

## Abstract

**Simple Summary:**

Due to the concern arising from the environmental impact caused by beef cattle, this study evaluated the greenhouse gas emissions of three production systems using increasing levels of pasture nitrogen fertilization during the backgrounding period. Following this, animals were finished either on pasture or feedlot. In addition, their contribution to human protein nutrient requirements was investigated. Both pastures without fertilization and with moderate fertilization resulted in the lowest greenhouse gas emission intensity. However, the number of animals increased twice, suggesting that moderate nitrogen fertilization enables the production of more meat using less area, which might contribute to decreasing deforestation. Moreover, tropical beef production grazing systems positively contributed to supplying the human protein requirements without competing with humans for food.

**Abstract:**

This study aimed to evaluate the greenhouse gas (GHG) emission and net protein contribution (*NPC*) of Nellore young bulls grazing marandu palisade grass (*Urochloa brizantha* cv. Marandu) under three levels of pasture nitrogen (N) fertilization during backgrounding and finished on pasture or feedlot, based on concepts of sustainable intensification. The treatments were: System 1: pastures without N fertilizer during backgrounding, and animals finished on pasture supplemented with high concentrate at a rate of (20 g of concentrate per kg of body weight; P0N + PS); System 2: pastures fertilized with 75 kg N ha^−1^ year^−1^ during backgrounding and animals finished on feedlot fed a total mixed ration (TMR; P75N + F); and System 3: pastures fertilized with 150 kg N ha^−1^ year^−1^ during backgrounding, and animals finished on feedlot fed a TMR (P150N + F). During backgrounding, all pastures were managed under a continuous and put-and-take stock grazing system. All animals were supplemented with only human-inedible feed. Primary data from systems 1, 2 and 3, respectively, in the field experiment were used to model GHG emissions and *NPC* (a feed-food competitiveness index), considering the backgrounding and finishing phases of the beef cattle production system. Average daily gain (ADG) was 33% greater for the N fertilizer pastures, while carcass production and stocking rate (SR) more than doubled (P75N + F and P150N + F). Otherwise, the lowest GHG emission intensity (kg CO_2_e kg carcass^−1^) was from the P0N + PS system (without N fertilizer) but did not differ from the P75N + F system (*p* > 0.05; pastures with 75 kg N ha^−1^). The main source of GHG emission in all production systems was from enteric methane. Moreover, *NPC* was above 1 for all production systems, indicating that intensified systems contributed positively to supply human protein requirements. Moderate N fertilization of pastures increased the SR twofold without increasing greenhouse gas emissions intensity. Furthermore, tropical beef production systems are net contributors to the human protein supply without competing for food, playing a pivotal role in the food security agenda.

## 1. Introduction

Grasslands comprise around 26% of Earth’s terrestrial land and livestock take over two-thirds of global agricultural land. Much of this land is not suitable for growing crops, and it is composed of rangeland that is too rocky, steep, or arid to support cultivated agriculture, yet this land can support cattle and protein upcycling [1]. Most grasslands are in tropical developing countries, where they are essential to the economic and social panorama and food supply of these regions [2]. However, tropical beef cattle production has lately been the target of a major negative impact from public opinion due to the low technological inputs in the production systems, which can directly impact the productive indexes, deforestation process, food competition, and the emission of greenhouse gases (GHG).

In livestock, the main sources of GHG emission are associated with enteric methane (CH_4_) from rumen fermentation, CH_4_ from feces, direct and indirect nitrous oxide (N_2_O) from animal excreta and fertilizer, and carbon dioxide (CO_2_) from deforestation, biomass burning, energy usage and fossil fuels [3].

In this sense, the sustainable intensification of tropical pastures presents itself as an essential tool, which aims to increase the productivity of the beef production chain, and simultaneously contribute to the gradual and proportional reduction of environmental impact [4]. Technologies such as grazing management, combined with fertilization and supplementation, are innovative strategies to improve animal performance, increase stocking rates and meat production per unit area, improve carcass and meat quality, and finally reduce the age at slaughter [5]. Although a source of N_2_O, the N fertilization of pastures has been shown to increase forage production, green leaf mass, pasture crude protein concentration and digestibility, resulting in higher stocking rates and productivity per area [6,7], which might contribute to prevent further deforestation because of lower land occupation factors. Associated with decreased deforestation, studies have shown that the GHG emission intensity (CO_2_e·kg^−1^ LW) decreased by 49.6 and 59.3% with increasing animal productivity [8,9].

Beyond their GHG emission, the environmental impact of beef production can be assessed by the ruminant’s ability to upcycle low-quality protein from by-products into high quality. Therefore, understanding the protein quality of beef relative to other protein sources in human diets is essential to understand the impacts of the beef value chain on human food supply. Beef products provide a more complete source of dietary protein (i.e., greater biological value) than plant sources, which contain insufficient levels of essential amino acids [10].

The dilemma at hand is how to meet the challenge of providing high-quality food to a growing human population while reducing the environmental footprint in a practical, economical, and timely fashion [4]. Thus, developing methods that accurately account for the GHG emissions using experimental field data and estimating the beef’s contribution to human nutrient supplies in tropical conditions are essential for addressing societal concerns and optimizing sustainability.

We hypothesized that livestock intensification, achieved by the fertilization of pastures combined with high supplementation and feedlot, will decrease the GHG emission intensity of tropical beef production due to the increase in productivity. Moreover, we demonstrated how the use of by-products for animal feed enhances the net protein contribution of the beef cattle chain to meet human protein requirements, leading to a reduced competition of livestock feed with human food. Therefore, we investigated the GHG emission intensity and net protein contribution of Nellore young bulls grazing marandu palisade grass (*Urochloa brizantha* cv. Marandu) under three levels of pasture nitrogen fertilization during backgrounding and finished on pasture or feedlot using by-products as concentrate ingredients.

## 2. Materials and Methods

### 2.1. System Boundaries

The boundaries of this study considered the backgrounding and finishing phases of the beef cattle production system using experimental field data. To calculate the GHG emission attributable on-farm agricultural activities, a life cycle inventory of GHG emissions resulting from animal activity (enteric CH_4_ and N_2_O and CH_4_ from manure), grassland nitrogen fertilization, diesel fuel use, and CO_2_ emissions from fuels and machinery was modeled, as described in Section 2.3. The functional unit was 1 kg of CO_2_ equivalent per kilogram of carcass (kg·CO_2_e·kg^−1^ carcass). The net protein contribution to the human food supply from beef cattle production was assessed according to Baber et al. [11]. Primary data from a field experiment were used to assess the effects of GHG emissions and net protein contribution. Primary data included animal and forage production, intake, diet digestibility, nitrogen balance, carcass production, and enteric CH_4_ emission during the backgrounding and finishing phases. 

### 2.2. Field Data Acquisition (Primary Data)

#### 2.2.1. Experimental Design and Treatments

The experiment was carried out in the Beef Cattle Center of São Paulo State University “Julio de Mesquita Filho”, Jaboticabal, SP, Brazil (21°15′2″ S 48°18′08″ W, 595 m altitude). The typical climate is humid subtropical, with dry winters and hot, wet summers. The experimental period involved two phases: backgrounding (wet season [December 2018 to May 2019]) and finishing (dry season [June to September 2019]). The average temperature throughout the experimental period was 22.4 ± 2.5 °C and the total precipitation was 927.4 mm (Appendix A). 

The animals used in this study were cared for according to the guidelines of the São Paulo State University Animal Care and Use Committee and the National Council of Animal Experimentation Control (protocol approval number 7979/18).

The treatments consisted of three beef cattle production systems with different levels and strategies of intensification, as follows: System 1: pasture without N fertilization during backgrounding followed by pasture grazing with high levels of supplementation during the finishing phase (P0N + PS); System 2: pasture with fertilization (75 kg N ha^−1^ year^−1^) during backgrounding followed by a total mixed ration (TMR) on feedlot during finishing (P75N + F); and System 3: pasture with fertilization (150 kg N ha^−1^ year^−1^) during backgrounding followed by a TMR on feedlot during finishing (P150N + F). During backgrounding, pastures from all systems were managed under a continuous and put-and-take stocking grazing system [12] to keep the canopy height at 25 cm [13]. Fertilized pastures received 75 or 150 kg N ha^−1^ as ammonium nitrate (32% N), which was divided into three similar doses. These three systems were selected because they represent common beef cattle practices in Brazil, whose main production system is grazing with different levels of intensification.

The experimental design was completely randomized, with three treatments and four replications (paddocks or stalls) per treatment, which was firstly determined based on the backgrounding phase that was predominantly in pastures. 

The experimental grazing site consisted of 24 hectares of marandu palisade grass (*Urochloa brizantha* Hochst ex A. Rich Stapf cv. Marandu) divided into 12 paddocks of approximately 2 hectares each, in a completely randomized design with four paddocks (replicates) per treatment. During backgrounding (172 days), all animals were managed under a continuous and put-and-take stock grazing system [12] to keep the canopy height at 25 cm [13]. Fertilized pastures received 75 or 150 kg N ha^−1^ as ammonium nitrate (32% N). The total fertilizer amount was fractioned in three applications of the same amount as follows: 19 December 2018 (25 and 50 kg N ha^−1^); 23 January 2019 (25 and 50 kg N ha^−1^) and 23 February 2019 (25 and 50 kg N ha^−1^), according to precipitation distribution.

Seventy-six 14-month-old Nellore bulls (Bos taurus indicus) were used, of which 48 (initial body weight (BW) 283 ± 15 kg) were tester animals (four animals per paddock) and the remaining 28 animals (put-and-take animals; initial BW 315 ± 35 kg) were used to maintain canopy height at 25 cm [13] with 28 days for each grazing cycle. During backgrounding, all animals received an ad libitum mineral mixture from December to March. Between April and June 2019, animals of all treatments were supplemented at a rate of 1 g per kg BW from a single supplement (Appendix A).

During the finishing phase (92 days), tester animals from the P0N + PS production system remained in the pasture and were supplemented at a rate of 20 g of concentrate per kg BW (Appendix A), while the animals from the P75N + F and P150N + F production systems were moved to a feedlot barn. During feedlot, the animals were kept in 10 × 6 m stalls (four tester animals per stall, the same four animals previously grouped from paddocks), and the diet was formulated with a roughage:concentrate ratio of 20:80 for daily gains of 1.2 kg d^−1^, as estimated by NRC [14]. The diets were formulated with human-inedible ingredients: sorghum silage, defatted corn germ, urea and mineral mix (Appendix A).

#### 2.2.2. Forage Mass and Morphological Composition

Forage samples were collected every 28 days during the entire grazing period in each paddock. The average canopy height was assessed every seven days, based on the measurement of 80 random points per paddock using a graduated ruler, to maintain the canopy height at 25 cm. At four average canopy height sites per paddock, forage mass was estimated after clipping all plants at soil level within the perimeter of a circular rim (0.25 m^2^). Samples were then separated into green leaves, dead material and stem and sheath and dried at 55 °C to a constant weight to estimate total forage dry matter (DM) per hectare in March 2019 (backgrounding phase) and August 2019 (finishing phase). For chemical composition analysis, additional forage samples were hand-plucked [15]. Data on canopy characteristics and chemical composition of the hand-plucked forage samples during the backgrounding phase, as well as data on the chemical composition of the hand-plucked forage samples during the finishing phase from the P0N + PS production system, are described in the Appendix A.

#### 2.2.3. Animal Performance

The average daily gain was measured in 48 tester animals (16 animals per treatment; 4 animals per paddock) by the difference between weight at the beginning and the end of each phase after a 14 h feed and water withdrawal. In addition, animals were weighed every 28 days (without previous fasting) to calculate the stocking rate, total herbage allowance, and the BW gain per hectare (GPH). The stocking rate was calculated based on the number of bulls in each paddock (sum of testers and put-and-take bulls) and their weight. The animal unit (AU) used in this evaluation was considered 450 kg BW [16].

#### 2.2.4. Intake, Digestibility, Nitrogen Balance and Enteric Methane

Intake, digestibility, nitrogen balance, and enteric methane measurements were performed in eight tester animals per treatment (two per paddock). Two evaluations were performed on the same animal: one during the backgrounding phase (March 2019) and the other during the finishing phase (August 2019). Intake and digestibility of forage or TMR diet were estimated based on data on fecal production and the indigestible neutral detergent insoluble fiber (iNDF) as an internal marker for assessing pasture DM intake. For the estimation of fecal output, chromium oxide (Cr_2_O_3_) was supplied orally at a dose of 10 g (packaged in paper cartridges) for ten days, of which the first seven were used for adaptation and the final three for the collection of feces, with three collections per day. Feces spot samples were collected from the ground after spontaneous defecation at three different times (7:00 a.m., 11:00 a.m., and 4:00 p.m.).

Hand-plucked forage samples were used to represent the diet (forage) consumed by grazing animals in each paddock [15] during each phase. The chemical composition of the hand-plucked forage is depicted in the Appendix A. The iNDF content of forage and fecal samples were estimated according to Valente et al. [17]. Forage intake was calculated as described by San Vito et al. [18]. The offered feed and orts were recorded daily for individual stalls and sampled twice a week for feedlot animals. Urine collections were performed in the form of a spot sample during spontaneous urination in the morning for three consecutive days. Daily urinary volume was estimated based on creatinine excretion, as described previously [19]. Approximately 40 mL of pure urine was separated for analysis of total nitrogen.

To evaluate enteric methane emissions, the sulfur hexafluoride (SF_6_) marker technique was adopted. A permeation capsule containing SF_6_ with a known release rate was inserted into the animal rumen [20]. A halter equipped with a capillary tube was fitted to the animal’s head, with prior adaptation, and connected to a PVC (polyvinyl chloride) chamber equipped with a valve and register, previously subjected to vacuum. After sampling, the yoke was pressurized with N, and the concentrations of CH_4_ and SF_6_ were determined by gas chromatography (Shimadzu GC2014 Gas Chromatograph, Kyoto, Japan), fitted with an electron capture detector (350 °C) to determine SF_6_, and a flame-ionization detector (250 °C) to determine CH_4_ concentration. The gas chromatograph was fitted with a 3.3 m molecular sieve column with an i.d. of 0.32 mm and film thickness of 300 μm (Alltech Associates, Auckland, New Zealand). The column and injector temperatures were both 85 °C but baked out at 200 °C daily. Nitrogen was used as the carrier gas at a flow rate of 40 mL/min. The measurements were performed on five consecutive days in each phase (backgrounding and finishing). Daily enteric CH_4_ emission was calculated as previously described [21].

#### 2.2.5. Chemical Analyses

Samples of hand-plucked forage, feed ingredients, orts, and feces were dried at 55 °C for 72 h in a forced-air oven for DM determination and ground in a Wiley mill (Arthur H. Thomas Co., Philadelphia, PA, USA) through a 1 mm and 2 mm screen. Dried and 1 mm ground samples were analyzed for DM (method 934.01; [22]), ash (method 942.05; [22]), total N content (FP, Leco Instruments Inc., St. Joseph, MI, USA), and neutral detergent fiber exclusive of ash (NDFom) (Ankon 200 fiber analyser, Ankom Technologies, Fairport, NY, USA). Dried and 2 mm ground samples of forage and feces were analyzed for iNDF content, according to Valente et al. [17]. Urine was analyzed for DM and N content (method 978.02; [22]).

#### 2.2.6. Slaughter and Carcass Evaluation

At the end of the finishing phase, all animals were slaughtered in a commercial slaughterhouse following standard industry procedures. The carcass of each animal was divided into two half-carcasses, which were weighed to determine the weight of the hot carcass and then chilled in a cold room at 0 °C for 24 h. 

### 2.3. Greenhouse Gas Emission

To assess the environmental impact, a life-cycle assessment approach was chosen, considering only GHG emissions. System boundaries were limited to the backgrounding and finishing phases, with a 264-day timeframe. Greenhouse gas emission was calculated as CH_4_ production from enteric fermentation and cattle dung; N_2_O emission from manure and urine deposited in pastures or stalls; N_2_O emissions from field fertilization; and fossil CO_2_ emissions from animal feed and fertilizer production, manufacturing, transportation, diesel use for farm operations, and power generation. Emissions related to buildings and machinery, veterinary and pesticide products, and emissions beyond the farm gate (such as transportation to slaughterhouses and carcass processing) were not included in the analysis.

Soil organic carbon was assumed to be at equilibrium across all production systems. The GHG emissions from each treatment were calculated using the Intergovernmental Panel on Climate Change (IPCC) [3] Tier 2 methodologies.

Regional emission factors (EF) were used in this study when available. Enteric CH_4_ emission was modeled using Tier 2 refinement methods of IPCC [3]. Average daily enteric CH_4_ emission measured using the SF_6_ tracer gas was considered. A Brazilian study conducted at the same site estimated CH_4_ emissions from manure deposited on pastures to be 0.54 kg CH_4_·head^−1^·year^−1^ [23]. Methane emission from feces in the feedlot was determined based on the IPCC [3] 10.23 equation, considering the volatile solid excretion from the measured intake and digestibility, and CH_4_ conversion factor of 2% for a dry lot in warm climate conditions (Table 10.17; [3]).

During backgrounding and finishing, N excreted through feces and urine was measured in the field. During backgrounding, direct emissions of N_2_O from feces and urine excreted on the pasture were estimated separately using N_2_O–N EFs (EF3PRP) for grazing beef cattle (0.36% and 1.02%, respectively) based on a Brazilian study [23]. According to a Brazilian study, during finishing, direct N_2_O emission (EF3PRP) from manure in the P0N + PS production system was 0.34% [23]. For feedlot animals (P75N + F and P150N + F production systems), the standard EF recommended in IPCC Chapter 10 was used to estimate direct N_2_O emissions (Table 10.21, [3]). For both phases and systems, indirect N_2_O emission was estimated based on EFs recommended in chapter 11 (Table 11.3, [3]), except for the fractions of total N from animal manure and urine emitted as NH_3_ (FracGASM) in the pasture during backgrounding and finishing (6.4% and 11.5%, respectively; [23].

The GHG emissions from fuel, electricity, fertilizers, and feeds used in the production systems were accounted for using the IPCC factors and other sources identified and cited in the literature [8,24]. The EFs used and assumptions for each source and input are described in Table 1. The fossil fuel requirements from the sorghum silage were taken from an updated version of Cardoso’s spreadsheet [8] using inputs and yields from actual management in the field. The sorghum was sown in November 2018 in a 2 ha area under no-till. In February 2019, whole-crop sorghum was harvested with yields of 41.88 tons of fresh silage ha^−1^.

Greenhouse gas emission was assessed according to the model and methodology developed by the IPCC [25]. Emissions were assigned as a function of carcass weight (kg·CO_2_e·kg^−1^ carcass). All data have been converted to their 100-year global warming potential in CO_2_e. The CO_2_e values for CH_4_ and N_2_O are 28 and 265, respectively [25].

**Table 1 animals-12-03173-t001:** Emission factors of purchased resources and feeds.

Source	Unit	Value	Reference
Diesel fuel use	kg·CO_2_e·L^−1^	3.53	[8]
Electricity	kg·CO_2_e·MWh^−1^	115	EPE
Lime	kg·CO_2_e·kg^−1^	0.48	[8]
Fertilizer			
Nitrogen (ammonium nitrate)	kg·CO_2_e·kg^−1^	5.50	[26]
Purchased feed			
Mixed mineral	kg·CO_2_e·kg^−1^	0.16	[8]
Cottonseed meal	kg·CO_2_e·kg^−1^	0.91	Feedprint ^1^
Defatted corn germ meal	kg·CO_2_e·kg^−1^	0.27	Feedprint ^1^

^1^ [24]. http://webapplicaties.wur.nl/software/feedprintNL/ (accessed on 16 March 2021).

### 2.4. Beef’s Contribution to Meeting Human Protein Requirements

A summative model of *NPC* was used to estimate beef’s contribution to meeting human protein requirements. System boundaries were limited to the backgraunding and finishing phases of this study, with a timeline of 264 days. The system approach and methodology described by Baber et al. [11] were used to estimate the *NPC* to the human food supply.

#### 2.4.1. Conversion Efficiency of Beef Cattle

Firstly, the human-edible protein produced (*HePp*) was calculated for each production phase (backgrounding and finishing) and the entire production system, based on the estimation of body protein excluded from the inedible fraction of empty body (Equation (2); [11]). To predict *HePp* for each size of the animal, body protein (*BP*) was estimated using empty body weight (*EBW*) that was estimated based on an equation for zebu cattle (Equation (1); [27]):(1)$$EBW=0.8507\times BW^{1.002}$$
where *EBW* is empty body weight in kg and *BW* is body weight in kg.
(2)$$HePp=\left(0.235\times EBW-0.00013\times EBW^{2}-2.418\right)\times \left(1-IBP\right)$$
where *HePp* is human-edible protein produced in kg, *EBW* is empty body weight in kg, and *IBP* is the proportion of inedible by-products, which represents 0.25 of *EBW* in steers [11,28]. The amount of *HePp* in the backgrounding and finishing phases was the difference between the final and initial *HePp*.

The human-edible protein of feed (*HePf*) represents the *HeP* removed from human food supply by the beef value chain. The *HePf* was estimated based on the sum of measured intakes of each ingredient multiplied by their edible portion (Table 2). Feedstuffs were classified as edible, partially edible, or inedible using criteria according to Wilkinson (2011) [29] and Ertl et al. (2016) [30]. *HePf* was summed across production phases to calculate the total *HePf* for the value chain.

The *HeP* conversion efficiency (*HePCE*; [30]) is a metric of comparison indicating the conversion of *HePf* into beef and was calculated using Equation (3):(3)$$HePCE=\frac{HePp}{HePf}$$
where *HePCE* is human-edible protein conversion efficiency in kg/kg, *HePp* is human-edible protein produced in kg, and *HePf* is human-edible protein of feed in kg. The entire production system’s *HePCE* was calculated as the sum of *HePp* from all phases divided by the sum of *HePf* from all phases.

#### 2.4.2. Assessing Protein Quality Using Digestible Indispensable Amino Acid Score

The assessment of protein quality of human-edible feedstuffs used in beef cattle diets was performed considering the ratio between the digestible indispensable amino acid score (DIAAS) of beef and that of the diet fed to cattle. The DIAAS was calculated with the ratio between mg of digestible indispensable amino acid in 1 g of dietary protein and mg of digestible indispensable amino acid in 1 g of reference protein, according to the Food and Agriculture Organization of the United Nations [31]. The digestible indispensable amino acid refers to any of the 10 indispensable amino acids.

The reference protein used in this model was the requirement published by the FAO (2011) [31] for children between the ages of 0.5 and 3 yr. When formulating diets for cattle, a weighted average of the DIAAS for human-edible feed ingredients was calculated for each amino acid. In this study, only feedstuff containing proteins potentially edible by humans was considered [30]. Amino acid composition and true ileal amino acid digestibility of each ingredient were obtained from the CVB Feed Table [32]. The smallest DIAAS for a single indispensable amino acid was assigned as the diet DIAAS on the premise of the first limiting amino acid (Table 2) and used to calculate the protein quality ratio (PQR).

The human-edible portion of the output product, e.g., beef, was 112, indicating that the amino acid profile of beef is superior to the requirements of a child (reference protein). To capture the change in the biological value of *HeP* that occurs when plant-derived *HeP* is converted to beef, PQR was calculated as the ratio between the DIASS of beef and DIASS of diet. A PQR was calculated for each phase of the beef production chain. To calculate the PQR regarding the entire period, the PQR was weighted based on the proportion of total *HePf* in each production phase.

#### 2.4.3. Net Protein Contribution

The *NPC* was calculated by multiplying the ratio of *HeP* in beef to the *HeP* in feedstuffs by the PQR (Equation (4)):(4)NPC=PQR×HePCE
where *NPC* is net protein contribution in kg/kg, *PQR* is protein quality ratio, and *HePCE* is human-edible protein conversion efficiency in kg/kg.

A *NPC* greater than 1 indicates that the value chain positively contributes to meeting human requirements. In contrast, an *NPC* less than 1 indicates the beef value chain is competing with humans for protein.

### 2.5. Statistical Analysis

Data variables were analyzed as a completely randomized design with three treatments (production systems) and four replicates, defined based on the backgrounding phase that was predominantly in pastures. The experimental unit was the paddock and, sequentially, the stall when applicable. The statistical model used was:$$Y_{i}:\mu +T_{i}+\varepsilon_{i},$$
where *Y* = the observed parameter, *T_i_* = treatment (2 degrees of freedom, df), and *ε* = the residual error associated with each observation as a random effect (9 df). Data were evaluated for homoscedasticity of variances and normality of errors. Residuals were plotted against the predicted values to validate model assumptions. Values with studentized residuals outside the ±2.5 range values were considered outliers. No outliers were identified. Means were compared using Tukey’s test and significance was declared as *p* ≤ 0.05. Statistical analysis was conducted using the MIXED procedure of SAS (version 9.4; SAS Institute, Cary, NC, USA).

## 3. Results

### 3.1. Individual Performance

The number of animals in each production system was determined by the stocking rate (SR) settled during backgrounding, in which SR of pastures fertilized with 75 and 150 kg N ha^−1^ year^−1^ (P75N + F and P150N + F production systems) were, on average, 78 and 118% greater, respectively, than those without N fertilization (P0N + PS system; Figure 1). Considering actual SR and mean shrunk body weight during backgrounding, the average number of animals per hectare increased from two (P0N + PS) to four and five young bulls in N-fertilized pastures (P75N + F and P150N + F beef cattle production systems, respectively). Therefore, the increase in N fertilization reduced the area required by each animal from 0.5 (P0N + PS) to 0.2 ha (P150N + F, Figure 1).

During backgrounding, ADG of animals from pastures fertilized with 75 and 150 kg N ha^−1^ year^−1^ (P75N + F and P150N + F systems) were, on average, 30 and 35% greater, respectively, than those from pastures without N fertilization (P0N + PS system; Table 3). Moreover, dry matter intake (DMI), DM digestibility, and individual enteric CH_4_ emission did not differ among systems (*p* > 0.05), whereas crude protein digestibility of animals from N-fertilized pastures (P75N + F and P150N + F systems) were on average 25% greater than those from pastures without fertilization (P0N + PS system; Table 3).

During finishing, DM and crude protein digestibility, ADG, and slaughter weight did not differ among systems (*p* > 0.05), but carcass weight of animals from feedlot (P75N + F and P150N + F systems) was on average 25% greater than those finished on pasture (P0N + PS system; Table 3). Although the DMI of animals from P0N + PS systems (pasture finished) was greater (*p* < 0.05), their individual enteric methane emission was lower than animals from P75N + F and P150N + F systems (*p* < 0.05; Table 3).

### 3.2. GHG Emission Intensity

During the separate backgrounding and finishing phases, carcass production from bulls of the more intensified systems (P75N + F and P150N + F) was higher compared to the P0N + PS system (*p* < 0.05; Table 4). Likewise, total GHG emissions of the more intensified systems were higher than the P0N + PS (*p* < 0.05; Table 4). Otherwise, GHG intensity (CO_2_e kg carcass^−1^) did not differ between the P75N + F system and the other systems (*p* > 0.05), but the P150N + F system presented the greatest emission (*p* < 0.05; Table 4).

Regarding the entire period, accounting for the backgrounding up to the finishing phase, carcass production from bulls of P75N + F and P150N + F doubled on average compared to the P0N + PS system (*p* < 0.05), whereas the total GHG emissions of the P75N + F and P150N + F systems increased 1.8 and 2.4 times more, respectively. Thus, the gain of the animals in the P150N + F system (more intensified) was not enough to offset the GHG emission intensity, since the emissions of CO_2_e kg carcass^−1^ of the P150N + F systems were on average 50% higher than the P0N + PS systems (Table 4). The GHG emission intensity of the P75N + F system, where pastures were fertilized with 75 kg N ha^−1^ year^−1^ and animals were finished in feedlot, did not differ from the other systems.

In general, the main source of emissions in all production systems was enteric CH_4_, contributing on average to 64% of total emissions (Figure 2). The second largest source of emission was manure, contributing in average to 12% of total emissions. Nonetheless, GHG emission intensity from manure, which was the sum of CH_4_ emission from feces and N_2_O emission from feces and urine deposited, was on average 74% higher (*p* = 0.005) in the P75N + F and P150N + F systems, where animals were finished in feedlot, compared to the P0N + PS system, where animals were finished on pasture. Feed emissions were similar among systems (*p* = 0.82). Emissions from fertilizer were null in the P0N + PS system but represented 12% and 18% of total emissions in the P75N + F and P150N + F systems, respectively (Figure 2). 

### 3.3. Beef’s Contribution to Meeting Human Protein Requirements

The DIAAS represents the human-edible protein quality of a food and its ability to meet the protein requirements of a child 0.5 to 3 yr of age. In this study, the sole human partially edible feedstuff was defatted corn germ, whose calculated DIAAS was low (34.8), resulting in a PQR of 3.22 (Table 5).

During backgrounding, *HePCE* (conversion of edible protein from feed into beef) was similar among the systems (*p* > 0.05; Table 5) and greater than 90. Likewise, *NPC* was also similar among the three systems (*p* > 0.05) and greater than 212, positively contributing to meet human protein requirements as indicated by *NPC* > 1 (Table 5).

During finishing, neither *HePCE* nor *NPC* differed among the systems (*p* > 0.05); however, *HePCE* was less than 1, indicating that the animals consumed more edible protein than produced, although the *NPC* greater than 1 indicated that the quality of protein offset it (Table 5).

Considering the entire period, *HePCE* was similar for all beef cattle production systems, with an average value of 1.19, indicating that the system produced 19% more protein than the edible protein consumed by the animal. Furthermore, the overall *NPC* was on average 3.6, indicating that the bulls produced 3.6 times more essential amino acids than they consumed, positively contributing to human protein requirements without competing with humans for *HeP* (Table 5).

## 4. Discussion

### 4.1. Performance and GHG Emission

The sustainable intensification of livestock systems has been pointed out as a suitable alternative to reduce the environmental impacts caused by meat production [33]. It has been shown that it is possible to increase livestock efficiency and mitigate GHG emissions through the adoption of technologies such as the use of fertilizers and adequate grazing management [6]. Higher forage production, greater cattle stocking rates, and greater carcass gain per animal can also reduce the need for land clearing, resulting in less deforestation [34]. In the present study, intensification of beef cattle production accounted for the increase in animal production per area due to moderate nitrogen fertilization. Thus, based on the sustainable intensification concept, this study evaluated the effect of intensification strategies on performance, GHG emissions, and net protein contribution of Nellore cattle grazing marandu palisade grass pasture under continuous stocking during backgrounding and finished on pasture or feedlot.

The greater carcass production of more intensified systems (P75N + F and P150N + F), mainly during the backgrounding phase when pastures were N-fertilized, resulted from the combination of the increase of animals per area and greater gain per animal. In this case, the intensification was achievable due to previous fertilization and adequate management of pastures. Those intensification strategies have the principle of manipulating soil–plant–animal factors, seeking a balance between supply and demand for food [6]. In the current study, the intensification via pasture N-fertilization resulted in greater forage mass production and nutritional value, which led to an increase in the stocking rate and gain per animal, likely because the animals harvested the high-quality leaf tissue before the forage entered senescence [35]. 

The greater number of animals per area, in the more intensified systems, resulted in more GHG production. However, the carcass gains achieved in the P150 + F system could not offset its GHG emissions, due to lower-than-expected weight gains and the high amount of fertilizer used (source of N_2_O emissions). In this regard, alternative nitrogen supply options to fertilizers might be promising technologies to mitigate N_2_O emissions while improving productivity [5]. For instance, a previous study observed that seed inoculation with mycorrhizal fungi, combined with the low rate of fertilizers, improved maize and sorghum silage without affecting forage nutritive value [36]. Future studies should focus on quantifying emissions from alternative sources of N.

Regarding the entire period, from backgrounding to farm gate, the environmental impact of the intensification systems, represented by the total GHG emissions, increased in the P75N + F and P150N + F systems. Although carcass production of the P150N + F system, where pastures were fertilized with 150 kg N ha^−1^ year^−1^ and animals were finished in feedlot, was higher than the P0N + PS system (pastures without N fertilization), during finishing, animals from all systems received the same concentrated diet, which provided a similar live weight gain and reduced the differences in gain achieved during backgrounding. As a result, the gain augments of P150N + F systems were not sufficient to offset the increase in GHG emissions, resulting in 50% higher emissions of kg CO_2_e kg carcass^−1^ than the P0N + PS. Conversely, GHG emission intensity of the P75N + F system, where pastures were fertilized with 75 kg N ha^−1^ year^−1^ and animals were finished in feedlot, was similar to the P0N + PS system (pastures without N fertilization), reflecting a balance between GHG emissions and carcass production. Indeed, previous studies have shown that the GHG emission intensity (kg CO_2_e kg carcass−^1^) decreased with increasing animal productivity [8,9,37]. It is worth noting that even the P0N + PS beef cattle production system of the present study showed some level of intensification through improvements in grazing management during backgrounding and by high concentrate supplementation during finishing.

To compare the results of this study with others that assessed the entire livestock life cycle carbon emission (from cradle to farm gate), the cow–calf phase was assumed to represent 50% of the total emission intensity of the beef cattle chain [8]. Therefore, the values in this study regarding the GHG emission intensity (kg CO_2_e kg carcass^−1^) of backgrounding plus finishing were doubled, resulting in 21.06, 28.26 and 31.52 kg CO_2_e kg carcass^−1^ for P0N + PS, P75N + F, and P150N + F systems, respectively, for the entire life cycle. Although the intensity of emissions increased with the intensification levels, these values are in accordance with the most intensified systems evaluated by Cardoso et al. [8], which considered five different scenarios, with increasing intensification levels, and total emissions were estimated at 58.3, 40.9, 29.6, 32.4, and 29.4 kg CO_2_e kg carcass^−1^ [8]. Since the first two scenarios cover less intensified systems than those evaluated in our study, this implies that the improvements in intensification achieved in the present study had a positive environmental impact. 

Another study [37] simulated different beef production systems and estimated CO_2_e emissions as 30.8, 34.4, 34.4, and 36.6 kg CO_2_e kg carcass^−1^ for fertilized pasture, re-seeded pasture, pasture with grain supplementation, and extensive pasture, respectively. Therefore, the simulated pasture improvements resulted in lower GHG footprints compared to the extensive system. However, more intensified systems emitted more GHG from land management (fertilization) and production of external resources (feed and fertilizer) compared to extensive [38]. Dick et al. [9] observed even greater benefits in intensified systems, as livestock emissions reduced from 45 to 18.32 kg CO_2_e kg carcass^−1^; however, in Dick et al.’s [8] study, the grazing systems were not fertilized with N, and the animals did not receive concentrates in any phase of their life cycle.

The SR of the pastures without N fertilization (P0N + PS system) of our study was, on average, 1.76 AU ha^−1^, which is 66% greater than the Brazilian national average (1.06 LU; [39]) with the lowest GHG emission intensity, supporting the concept that improvements in grazing management per se are the most powerful tool to mitigate the environmental impact of grazing systems. Notwithstanding, pasture fertilization with 75 kg N ha^−1^ year^−1^ resulted in similar GHG emission intensity, but stocked twice as many animals than pastures without fertilization, suggesting that moderate N fertilization and improvements in grazing management are resourceful tools to move towards the sustainable intensification of livestock, enabling the production of more meat using less area, decreasing the need for deforestation. 

Our results indicate that CH_4_ from enteric fermentation is the largest proportion of total GHG emissions in contrasting beef cattle production systems, corroborating previous ruminant studies [40,41]. In our study, enteric fermentation responded to 64% of total CO_2_e emissions, a value higher than those found in studies conducted in Europe and Canada, where enteric CH_4_ emissions corresponded to 32 to 42% [37] and 48% [42] of total CO_2_e emissions. The lower contribution of enteric emissions obtained by these authors is a consequence of the greater contribution of emissions from manure management and animal production with large amounts of concentrates.

There were no observed differences in enteric CH_4_ emissions among the systems. However, the P0N + PS system presented an enteric CH_4_ contribution 15.8% higher than the intensified systems, once the latter had greater contribution from other sources related to fertilization [P75N + F (75 kg of N ha^−1^) and P150N + F (150 kg of N ha^−1^)]. The crucial issue regarding chemical fertilizers refers to their potential to harm the environment. In tropical pastures, ammonia (NH_3_) volatilization is one of the main loss pathways [43]. When applied to the soil, the nitrogen fertilizer rapidly undergoes hydrolysis by the action of the enzyme urease, releasing ammonia (NH_3_) and CO_2_ into the atmosphere [43]. In this study, the N fertilizer was ammonium nitrate, which contributed to approximately 11.6 and 18% of the total CO_2_e emissions in the entire period for the P75N + F and P150N + F systems, respectively.

Manure emissions were higher in the more intensified systems where animals were finished in feedlot. The manure produced by cattle can be a source of both CH_4_ and N_2_O emissions. Under grazing conditions, the excretions have little impact since the soil–plant system can use most of the nutrients present in the manure. In feedlots, due to the high concentration of animals, the large volume of feces and urine accumulated on the floor of the stalls can cause contamination through superficial carrying, leaching in the soil, or volatilization of gases.

In the present study, we estimated only the emissions from farm inputs and activities. The effect of carbon sequestration on reducing the associated CO_2_ emissions was not considered. Some authors have been giving attention to the potential of Urochloa pastures to accumulate soil C [44,45] because C sequestration can alter the GHG balance from emission sources to carbon sinks, especially when pastures are better fertilized [46]. A study comparing three contrasting production scenarios (degraded pasture, managed pasture, and integrated crop–livestock–forest system) in a palisade grass pasture in Brazil found that managed pastures fertilized with NPK can decrease the intensity of GHG emissions, due to C sequestration in the soil, in over 20% [40]. However, the sink for atmospheric CO_2_ does not increase indefinitely, and it is dependent on the input of organic material and its oxidation rate, the rate at which existing soil organic matter decomposes, soil texture, and climate [47]. Due to the absence of enough data available at the moment to use carbon sequestration as a CO_2_ mitigation strategy in different systems, this was not considered in this study. Future studies of site-specific data calculating soil carbon changes are needed for more specific grassland soil management practices recommendations to mitigate GHG emissions from livestock animals.

### 4.2. Beef’s Contribution to Meeting Human Protein Requirements

The *HePf* and *HePp* of the P0N + PS system were lower than the *HePf* of the other systems in the growing and finishing phases. This difference occurred because the *HeP* consumption and production calculations were per paddock, considering their respective stocking rates. As the paddock stocking rates increased with the intensification levels, the consumption and production of *HeP* increased concomitantly. Thus, *HePCE* was adopted as a comparison metric. Since in both the backgrounding and finishing phases, the animals from the different systems received the same diet (pasture-based backgrounding diet and concentrate-based finishing diet), *HePCE* did not differ among the systems.

The three systems had a similar *NPC* during the backgrounding and positively contributed to meeting human protein requirements, as indicated by *NPC* > 1. During this phase, all the animals received a pasture-based diet. A small amount of *HeP* (low-intake supplement) was incorporated and offered during the last three months of this phase, over the transition of the wet season to the dry season, when pasture suffers quantitative and qualitative losses. Around 90% of the supply of animal feed for total national meat produced is pasture [39], a product that is human-inedible. However, ruminants are efficient in the upcycling and conversion of pasture with low-quality proteins into beef, a high-quality protein source for humans [48].

During the finishing phase, the *HePCE* of all systems were below 1 (meaning during this phase, more *HeP* was being consumed than produced). However, the cattle’s ability to upcycle protein from low quality to high quality allowed an *NPC* greater than 1. In this way, the quality of the protein produced by the animals was superior to the protein consumed leading to a positive contribution to meet human protein requirements. In this study, a DIAAS was estimated for the diets fed and the human-edible portion of a beef carcass. The DIAAS (%) represents the ability of a human-edible foodstuff to meet the protein requirements of a child 0.5 to 3 years of age. The feedstuff offered for animal feed was considered partially edible and presented a DIAAS of 34, while the DIAAS of the animal protein produced was estimated at 112, proving much greater capacity of beef products to meet the human protein requirement.

A previous study carried by Baber et al. [11], comparing four different scenarios, embracing possible production parameters and industry diets used in the United States, found *NPC* values for the finishing phase to be much lower (0.30, 0.38, 0.84 and 1.07) than in the current study. Considering that ruminants are efficient at converting vast renewable resources, including by-products, into high-quality food that is edible for humans [49], this study used only diets comprised of by-products with low nutritional value for humans, which led to a small *HePf* when compared to conventional systems. In American conventional systems, large amounts of human-edible grains, such as corn, are fed to the animals, decreasing the *NPC*.

Considering the entire period, *HePCE* was similar for all beef cattle production systems, with an average value of 1.19. Although in the finishing phase, more *HeP* were being consumed than produced, it was outweighed by the backgrounding ability to positively contribute to the human food supply by using less *HeP* and improving the protein quality. When simulating different cattle-grazing scenarios in Brazil, a previous study [38] observed that the grazing system with grain supplementation (cows, heifers, and stockers fed with corn and soybean at a supplementation level to meet 45% of recommended protein requirements) presented a *HePCE* below 1, indicating a net reduction in potential human food driven by greater use of human-edible feed to produce animal products. These results reinforce the importance of using by-products as a viable strategy for future human food availability.

*NPC* for the entire period evaluated was above 1 for all systems, indicating each system was positively contributing to human protein requirements and were not competing with humans for *HeP*. Overall, the backgrounding had the greatest *NPC* when compared with the finishing phase, since during the backgrounding the main feed consumed by the animals was pasture, while in the finishing phase a concentrated supplementation was included.

Evaluating *NPC* of beef supply chains provides an additional piece of information to underpin a defensible life-cycle evaluation of ruminant production systems. In this study, we demonstrated that the concept of *NPC* is a useful sustainability metric to inform the food vs. feed debate. Additional methods for estimating climate impacts from beef production systems, as well as the occupation of land categories of arable vs. non-arable land, would be important aspects to include in future simulations of beef production systems.

## 5. Conclusions

The GHG emission intensity (CO_2_e kg carcass^−1^) of the systems where pastures are fertilized with 75 kg N ha^−1^ are like those without N fertilization, but stocking twice as many animals, suggesting that moderate N fertilization and improvements in grazing management are the most crucial technologies to mitigate the environmental impact of grazing systems. 

The beef value chain is a net contributor to the *HeP* available for human consumption. Furthermore, the quality of the *HeP* produced was enhanced throughout the beef value chain due to the ability of cattle to upcycle protein from low-quality to high-quality protein. Thus, it allowed all systems in both phases to have an average *NPC* of 3.59, positively contributing to human protein requirements without competing with humans for food.

## Figures and Tables

**Figure 1 animals-12-03173-f001:**
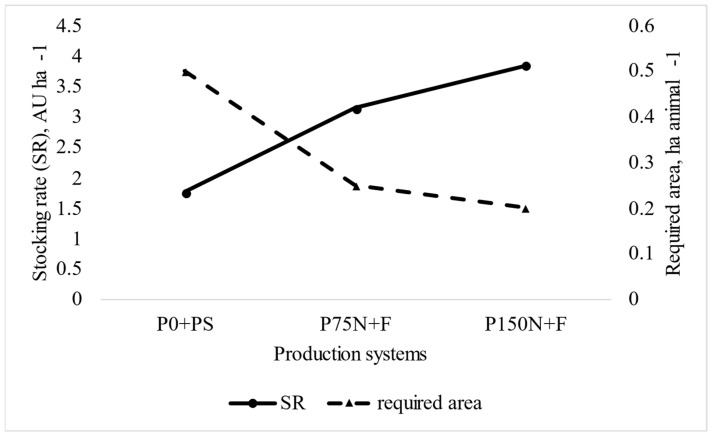
Stocking rate and required area per animal per animal in each production system.

**Figure 2 animals-12-03173-f002:**
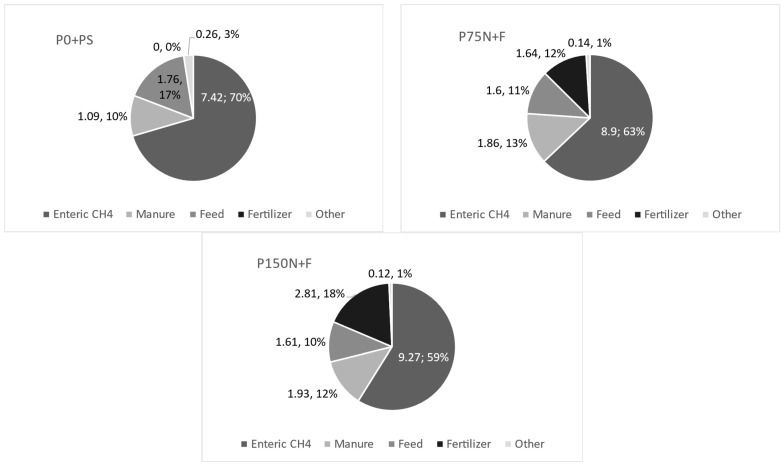
Sources of GHG emission in each production system.

**Table 2 animals-12-03173-t002:** Composition of the supplements and total mixed ration (TMR) fed to Nellore cattle across the backgrounding and finishing phases of all production systems.

Ingredient	Composition (%DM)				
	Supplement Backgrounding ^1^	Supplement Finishing ^2^	TMR ^3^	HEF ^4^	DIASS ^5^
Sorghum silage	-	-	20.0	0	
Defatted corn germ	21.9	94.4	75.5	0.5	34.8
Cotton seed meal	37.0	-	-	0	
Urea	4.6	1.8	1.5	0	
Mineral mix	36.5	3.8	3.0	0	

^1^ Fed to animals of all production systems from April to June during backgrounding phase. ^2^ Fed to animals of P0 + PS production system: managed pasture without N fertilization during backgrounding, and fed on pasture with high supplementation during finishing. ^3^ Fed to animals of P75 + F and P150 + F productions systems during finishing (P75N + F: managed pasture with fertilization of 75 kg N ha^−1^ year^−1^ during backgrounding, and fed a total mixed ration (TMR) on feedlot during finishing; and P150N + F: managed pasture with fertilization of 150 kg N ha^−1^ year^−1^ during backgrounding, and fed a TMR on feedlot during finishing). ^4^ Human-edible fraction: proportion of feed ingredients that is human edible. ^5^ Digestible indispensable amino acid score.

**Table 3 animals-12-03173-t003:** Performance, intake, digestibility, and methane production of Nellore yearling bulls in different production systems during the periods of backgrounding and finishing.

Production Systems ^1^					
Variable ^2^	P0N + PS	P75N + F	P150N + F	SEM	*p*
**Backgrounding**					
Days, d	172	172	172	-	-
Avg. initial SBW, kg	273	282	294	15.60	0.650
Avg. SBW at end, kg	385	423	437	20.00	0.210
Avg. daily gain, kg d^−1^	0.62 ^b^	0.81 ^a^	0.84 ^a^	0.030	0.001
Dry matter intake, %BW	2.4	2.5	2.5	0.230	0.970
Dry matter digestibility, %	50.5	55.3	50.1	2.910	0.410
Crude protein digestibility, %	45.2 ^b^	57.3 ^a^	55.9 ^a^	2.870	0.028
Individual enteric methane, g d^−1^	141.2	173.3	179.6	16.360	0.260
**Finishing**					
Days, d					
Avg. daily gain, kg d^−1^	0.92	1.01	1	0.074	0.640
Dry matter intake, %BW	2.8 ^a^	2.4 ^a,b^	2.3 ^b^	0.130	0.046
Dry matter digestibility, %	58.7	61.9	60.2	1.820	0.450
Protein digestibility, %	61	63.8	60.7	4.250	0.820
Individual enteric methane, g d^−1^	135.0 ^b^	218.6 ^a^	224.5 ^a^	22.20	0.033
Avg. slaughter SBW, kg	460	504	510	17.10	0.130
Carcass weight, kg	265 ^b^	300 ^a^	304 ^a^	10.80	0.055
Dressing percentage, %	59.9	59.3	59.4	0.600	0.750

^1^ P0 + PS: pasture without N fertilization during backgrounding and fed on pasture with high supplementation during finishing; P75N + F: pasture with fertilization of 75 kg N ha^−1^ year^−1^ during backgrounding and fed a total mixed ration (TMR) on feedlot during finishing; and P150N + F: pasture with fertilization of 150 kg N ha^−1^ year^−1^ during backgrounding and fed a TMR on feedlot during finishing. ^2^ Avg. = average; BW = body weight; SBW = shrunk body weight. Within rows, means followed by different letters indicate statistical differences (*p* < 0.05).

**Table 4 animals-12-03173-t004:** Greenhouse gas emissions of Nellore yearling bulls in different production systems during the periods of backgrounding and finishing and the entire period.

Production Systems ^1^					
Variable ^2^	P0N + PS	P75N + F	P150N + F	SEM	*p*
**Backgrounding**					
GHG total, kg CO_2_e ha^−1^	1994 ^b^	4634 ^a^	6134 ^a^	397	0.001
Carcass production, kg ha^−1^	134 ^b^	272 ^a^	311 ^a^	21.1	0.001
GHG intensity, kg CO_2_e kg carcass^−1^	5.87 ^b^	7.32 ^a,b^	8.35 ^a^	0.522	0.025
**Finishing**					
GHG total, kg CO_2_e ha^−1^	1580 ^b^	4429 ^a^	5363 ^a^	582	0.0034
Carcass production, kg ha^−1^	203 ^b^	366 ^a^	423 ^a^	35.8	0.005
GHG intensity, kg CO_2_e kg carcass^−1^	4.65 ^b^	6.80 ^a,b^	7.40 ^a^	0.652	0.036
**Entire period**					
GHG total, kg CO_2_e ha^−1^	3574 ^b^	9063 ^a^	11,498 ^a^	838	0.0003
Carcass production, kg ha^−1^	337 ^b^	637 ^a^	734 ^a^	45.8	0.0004
GHG intensity, kg CO_2_e kg carcass^−1^	10.53 ^b^	14.13 ^a,b^	15.76 ^a^	0.915	0.008

^1^ P0 + PS: pasture without N fertilization during backgrounding and fed on pasture with high supplementation during finishing; P75N + F: pasture with fertilization of 75 kg N ha^−1^ year^−1^ during backgrounding and fed a total mixed ration (TMR) on feedlot during finishing; and P150N + F: pasture with fertilization of 150 kg N ha^−1^ year^−1^ during backgrounding and fed a TMR on feedlot during finishing. ^2^ GHG = greenhouse gases; CO_2_e = CO_2_ equivalent, CH_4_ = methane. Within rows, means followed by different letters indicate statistical differences (*p* < 0.05).

**Table 5 animals-12-03173-t005:** Net protein contribution of Nellore yearling bulls in different production systems, during the periods of backgrounding and finishing and the entire period.

	Production Systems ^1^				
Variable ^2^	P0N + PS	P75N + F	P150N + F	SEM	*p*
**Backgrounding**					
Diet DIAAS	34.8	34.8	34.8		
PQR	3.2	3.22	3.22		
Total *HePf*, kg paddock^−1^	0.6 ^b^	1.16 ^a^	1.31 ^a^	0.11	0.004
Total *HePp*, kg paddock^−1^	55.2 ^b^	118.3 ^a^	129.4 ^a^	8.53	<0.001
*HePCE*	90.8	103.7	99.4	5.06	0.24
*NPC*	212.7	242.7	232.7	11.84	0.24
**Finishing**					
Diet DIAAS	34.8	34.8	34.8		
PQR	3.2	3.22	3.22		
Total *HePf*, kg paddock^−1^	98.9	162.1	162.9	23.58	0.14
Total *HePp*, kg paddock^−1^	36.7 ^b^	68.6 ^a^	70.2 ^a^	7.47	0.019
*HePCE*	0.4	0.5	0.5	0.059	0.51
*NPC*	1.1	1.4	1.4	0.18	0.51
**Entire period**					
Diet DIAAS	34.8	34.8	34.8		
PQR	3.22	3.22	3.22		
Total *HePf*, kg paddock^−1^	99.5	163.2	164.2	23.6	0.14
Total *HePp*, kg paddock^−1^	91.9 ^b^	186.9 ^a^	199.6 ^a^	12.8	<0.001
*HePCE*	0.92	1.31	1.34	0.24	0.45
*NPC*	2.79	3.95	4.04	0.74	0.44

^1^ P0 + PS: pasture without fertilization during backgrounding and finishing on pasture with high supplementation; P75N + F: pasture with fertilization of 75 kg nitrogen (N) ha^−1^ year^−1^ during backgrounding and finishing on feedlot; and P150N + F: pasture with fertilization of 150 kg nitrogen (N) ha^−1^ year^−1^ during backgrounding and finishing on a feedlot. ^2^ DIAAS = digestible indispensable amino acid score (%), PQR = protein quality ratio, *HePf* = human-edible protein consumed, *HePp* = human-edible protein produced, *HePCE* = human-edible protein conversion efficiency, *NPC* = net protein contribution. Within row, means followed by different letters indicate statistical differences (*p* < 0.05).

## Data Availability

The data presented in this study are fully available in this article and the Appendix A.

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
