# Peer review of "Response of Pasture Nitrogen Fertilization on Greenhouse Gas Emission and Net Protein Contribution of Nellore Young Bulls"

_animals, 2022, doi:10.3390/ani12223173_

Round 1

Reviewer 1 Report

Your simple summary is not representing any result of GHG emission. Please include it

Line 14-18: too long sentence, please split it into two sentences

‘Due to the concern arising from the environmental impacts caused by beef cattle, this study evaluated the greenhouse gas emissions of beef cattle production systems and their contribution to attend the human protein nutrient requirements, under three levels of pasture nitrogen fertilization during the backgrounding and finished on pasture or feedlot, based on concepts of sustainable intensification.’

Line 18-19: what do you mean by ‘N’ in sentence. Term should be presented full before being abbreviated

‘Our results suggest that the increase in animals per area supported by moderate N fertilization of pastures enables the production of more meat using less area……’

Line 27: not clear ‘…animals high supplemented on…’

Line 33: ‘…per kg BW’ BW term should be appeared full on first seen. As it is not using in subsequent sentences of abstract therefore, no need to write this term

Line 38: is this statistically different ? if yes, please use P values where needed ‘..but did not differ from P75N+F system (pastures with 75 kg N ha-1 )…..’

Line 39: Remove CH4 from the sentence ‘Enteric methane (CH4) was the main contributor…’

Line 39: Rewrite these results ‘Enteric methane (CH4) was the main contributor….’ Its not clearing GHG emissions and other gases

Line 41-42: is this your results ‘Improvements in grazing management stands for the most crucial technology to mitigate the environmental impact of tropical beef grazing systems….’ Your study is not grazing management instead N fertilization of pasture etc..

Line 42-44: seems illogical sentence here please ‘Furthermore, tropical beef production systems are net contributors to the human protein supply without competing for food, playing a pivotal role in the sustainability of tropical beef production’

Line 58-60: please include suitable references ‘In livestock, the main sources of GHG emission are associated with enteric CH4 from rumen fermentation, CH4 from feces, direct and indirect N2O from animal excreta and fertilizer, and CO2 from deforestation, biomass burn, energy usage and fossil fuels’

Line 58-60: please provide full name of abbreviation used at first seen ‘In livestock, the main sources of GHG emission are associated with enteric CH4 from rumen fermentation, CH4 from feces, direct and indirect N2O from animal excreta and fertilizer, and CO2 from deforestation, biomass burn, energy usage and fossil fuels’

Line 83-84: what do you mean by ‘To properly evaluate animal production, humanly consumable energy and protein intake should also be used for efficiency comparisons’ did you account above mentioned parameters in current study

Line 156: as as estimated by? Please correct reference

Line 193-195: correct the reference in the sentence ‘The iNDF content of forage and fecal samples were estimated according to Valente et al. (2011)[16]. Forage intake was calculated as described by San Vito et al., 2016 [17].’

Line 205-206: ‘After sampling, the yoke is pressurized with N…’ yoke is or yoke was

Line 212: DM isn’t abbreviated before

Line 226-227: please avoid the use of we, I in the sentence ‘To assess the environmental impact, we adopted a life cycle assessment approach in which only GHG emissions were considered.’

Line 228: don’t start sentence with abbreviation

Line 267: don’t start sentence with abbreviation

Line 277: correct citation ‘……………… described by [10] were used to estimate the NPC to the human food supply’

Line 363: Remove ‘SBW’ from sentence ‘…sidering actual SR and mean shrunk body weight (SBW) during backgrounding’ because it is not being used in the subsequent section

Line 370: what do you mean by ADG in the sentence ‘During backgrounding, ADG of animals from pastures fertilized with 75 and 150 kg’ term should be appeared on first seen please

Line 373: dry matter has already been abbreviated in M&M. please don’t abbreviate again

Line 373: correct ‘Moreover, dry matter (DM) intake (DMI)’ DMI

Line 373: why are you not using methane abbreviation instead of term. You have already abbreviated it

Line 502-503: how did you considered ‘we considered that the cow-calf phase represents 50% of the total emission intensity of the beef cattle chain’ rewrite the sentence please

Line 503: would be or was ?

Line 512-514: what do you mean by this sentence ‘Another study simulated different beef production systems and estimated CO2e  emissions as 30.8, 34.4, 34.4, and 36.6 kg CO2e kg carcass-1 for pasture fertilized, pasture re-seeded, pasture with grain supplementation, and extensive pasture, respectively.’ do you refer other study? If yes, please give reference

Line 516: what ‘they’ referd in the sentence? Please be specific

Line 519: correct the reference Dick´s

Line 607-608: this was aim of your study? ‘we aimed to explore the use of by-products, with low nutritional value for humans, into the cattle’s diet, which led to a small HePf when compared to conventional systems’

Reviewer 2 Report

Dear authors,
I reviewed the manuscript identified as animals-1943765 (Response of pasture nitrogen fertilization on greenhouse gas emission and net protein contribution of Nellore young bulls). The works of this type are extremely interesting and necessary to enhance the beef production system's sustainability, although, according to many, they are not innovative. Congratulations on your effort. As a recommendation, I suggest that the authors correctly check the appropriateness of the term "backgrounding", which is widely used throughout the text. I understood it as "basic fertilization". In addition, although the problem is correctly described, the authors could also refer, for example in the introduction, to alternative nitrogen supply options to fertilizers. In this regard, the authors could refer to doi.org/10.1017/S0021859618000072. I believe that this aspect, at least as a perspective, can be usefully recalled.
I give you my best wishes

Reviewer 3 Report

Dear authors, it seems to me that this manuscript has great relevance in the scientific world. However, some important points affect the quality of the manuscript.

 General comments:

 Material and methods:

Dear authors, your manuscript is well written. I like the writing style of the topics. However, the topic of Material and Methods is confusing to me. I think that in order to improve this topic; you should add a subtopic "Experimental Design" to specifically describe the design of the current experiment. I found parts of the design in many different paragraphs, and this made it difficult to identify the specific experimental design.

Furthermore, due to the low number of observations, I believe that adding the statistical model and a good description of the experimental design will help to better understand the results and estimate if the number of observations was sufficient to find correct results or if the power of the test it wasn't enough.

 Specific comments:

 Lines 24-27: The objective in lines 24-26 is well written; however, when you try to enter the production system, there is a bit of confusion. Please rewrite the objective trying to fit line 27. Maybe something like: “… from Nellore bulls under feedlot or grazing marandú … managed in three levels of nitrogen fertilization”

 Lines 28-30: Rewrite it. E.g.: Grazing bulls (unfertilized pastures) finished with a high concentrate rate (20 g concentrate/kg body weight) … or something less confusing.

 Lines 37-41: Greater or lower is a good answer; however, a number is better. Add numbers. E.g. is 10% greater than ….

 Lines 42-46: These lines are more like comments than a conclusion. Rewrite it.

 Line 98: You describe sometimes about the use of by-products. In this sense, I think that is necessary to complete this idea. E.g. … pasture or feedlot using by-products as concentrate ingredients.

 Line 104: This modeling is not mentioned or described in the statistical subtopic. Describe it in the statistical subtopic or cite the methodology used.

 Line 138: In lines 25 and 96 it is described as Marandu grass and here as palisade grass. Use a consistent name throughout the text. Perhaps inserting the various names in parentheses at the beginning might help.

 Lines 138-140: This part is very confusing. Why are the paddocks replicated? Shouldn't the replicas be the animals? Describe the correct experimental design in a new subtopic: Experimental design.

 Lines 143-144: … “which was divided into three similar doses.” Describe this better.

 Lines 145-148: What was the average weight of the 28 animals? What breed were the animals? What average age were the animals? Etc

 Line 146: Those 28 animals were considered in the statistical analysis? The experimental design needs to be rewritten to avoid being confusing.

 Lines 152-158: How many animals were used in each treatment? How many days were the animals on pasture and in feedlot?

 Line 156: Change “stall” instead of “pen” here and throughout the text.

 Line 178: Each phase? What phases? An experimental design subtopic is necessary.

 Lines 190-192: Total fecal collection or spot collection? Collection of feces from the rectal ampulla or soil?

 Lines 196-197: I assume this is a recommendation from another reviewer; however, change “et al.” in place of “and collaborators” here and throughout the text.

 Line 209: Describe better this part. The use of temperature, standard, column, etc.

 Lines 224-226: Was only the weight of the hot carcass measured?

 Lines 351-358: Please, for a better understanding, describe the statistical models used.

 Line 470: Palisade or marandu? I know you can use both names; however, homogenizing is better.

 Lines 645-647: Change the position of these lines, because this is a comment from the authors. My suggestion is to place it after the following idea “…technology to mitigate the environmental impact of grazing systems.”

 Line 655: Use the current value found or an average value instead of "greater than 1" to describe the importance of NPC.

Round 2

Reviewer 1 Report

Authors have sufficiently improved the manuscript 

Author Response

We are thankful to the time that the reviewer of this Journal dedicated to this manuscript

Reviewer 3 Report

Dear authors, I congratulate you on the work you have done.  You made corrections to the manuscript according to my suggestions.  I agree with the changes and believe the manuscript can be published as is.